# Personalized Federated Recommendation for Cold-Start Users via Adaptive Knowledge Fusion

## Abstract

Federated Recommendation System (FRS) usually offers recommendation services for users while keeping their data locally to ensure privacy. Currently, most FRS literature assumes that fixed users participate in federated training with personal IoT devices (e.g., mobile phones and PC). However, users may join incrementally, and retraining the entire FRS for each new participating user is unfeasible due to the high training costs and the limited global knowledge contribution from a small number of new users. To guarantee the quality service for these new users, we take a dive into the federated recommendation for cold-start users, a novel scenario where the new participating users can directly obtain a promising recommendation without comprehensive training with all participating users by leveraging both transferred knowledge from the converged warm clients and the knowledge learned from the local data.

Nevertheless, the efficient transfer of knowledge from warm clients remains controversial. On the one hand, cold clients may introduce new sparse items, resulting in a shift in the item embedding distribution compared to that converged on warm clients. On the other hand, cold-start users need to match similar user information from warm clients for a collaborative recommendation, but directly sharing user information is a violation of privacy and unacceptable. To tackle these challenges, we propose an efficient and privacy-enhanced federated recommendation for cold-start users (FR-CSU) that each client can adaptively transfer both user and item knowledge separately from warm clients and implement recommendations with local and transferred knowledge fusion. Specifically, each cold client will train a mapping function locally to transfer the aligned item embedding. Meanwhile, warm clients will maintain a user prototype network collaboratively that provides privacy-friendly yet effective user information for cold-start users. Then, a linear function system will integrate the transferred and local knowledge to improve recommendations. Extensive experiments show that FR-CSU achieves superior performance compared to state-of-the-art methods.

## CCS Concepts

• **Computing methodologies → Multi-agent systems**; **Knowledge representation and reasoning**.

## Keywords

Federated Learning, Recommendation System, Cold-Start User.

*WWW'25, 28 April - 2 May, 2025, Sydney, Australia*

© 2018 Copyright held by the owner/author(s). Publication rights licensed to ACM.
ACM ISBN 978-1-4503-XXXX-X/18/06
https://doi.org/XXXXXXX.XXXXXXX

## 1 Introduction

Recommendation systems have become essential tools and products, profoundly influencing daily lives by providing personalized suggestions for items that may interest users. These systems primarily depend on centralized servers to gather user characteristics, item features, and preferences to train models for accurate recommendations [11, 35, 39]. However, transmitting local user information to central servers raises significant privacy and security issues. Additionally, recent strict government regulations on privacy protection, such as GDPR, emphasize the importance of storing user data locally on devices instead of uploading it to a central server. To address this challenge, federated learning (FL) has emerged as a promising solution, facilitating data localization and enabling the distributed training of a globally shared model [14, 23, 30, 31]. FL alternates between local model training on client devices and aggregation of these models on the server. This framework has achieved great success and has been applied in various domains, including recommendation systems [10, 25, 27, 41] and smart healthcare [5, 6].

In recent years, researchers have focused on federated recommendation systems (FRS) that provide optimal recommendations for users without compromising data privacy. The first FL-based collaborative filtering method, FCF, introduced in [1], uses the stochastic gradient approach to update the global model with the FedAvg algorithm. The study in [3] adapts distributed matrix factorization to the FL environment, incorporating homomorphic encryption on gradients before uploading to the server. FedNCF [12] applies neural collaborative filtering in a federated context, leveraging neural networks to learn user-item interactions and enhance model learning. To strengthen user privacy protection, [21] proposes a dual personalization mechanism for personalized recommendations by keeping user embeddings local. Building on this, FedRAP [15] distinguishes the differences between different clients through additive personalization techniques.

While these methods achieve remarkable success in the federated recommendation, they assume that the number of participating users in the system is fixed and static. However, in a realistic federated recommendation application, new users may emerge and request model updates from the server to obtain satisfactory recommendations. The illustration of this scenario can be found in a simple example in Figure 1. In traditional federated learning scenarios, this is often defined as a federated continual learning task, where the system needs to incorporate new tasks into the original ones and retrain an optimal global model. However, such a training mode is impractical for a federated recommendation for four reasons: (1) Clients in the federated recommendation are typically IoT devices of different users, and the number of clients is often significant. Retraining the entire federated recommendation system would incur significant training and communication costs. (2) Retraining may affect the already converged global model, degrading the recommendation performance for previous users. (3) Previous users may not be

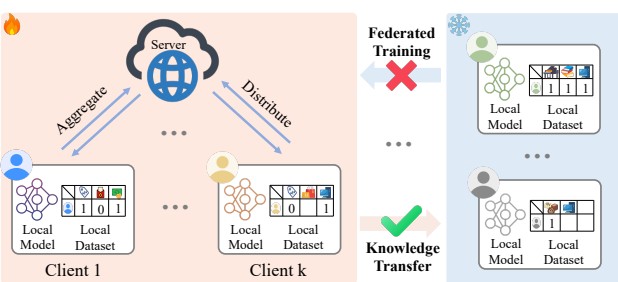

(a) Learning on warm clients      (b) Inference on cold clients

**Figure 1: The illustration of the federated recommendation for cold-start users. During the training phase of warm clients, all clients train on their local data and exchange knowledge through federated aggregation on the server. When cold-start users arrive, they will no longer participate in federated training and can only improve their local recommendation performance by utilizing their local knowledge and partially transferring knowledge from the warm clients.**

willing to participate in retraining, as the number of new incremental users is relatively small compared to the previous users, resulting in negligible recommendation performance gains. (4) If the server accumulates sufficient new users for retraining, these new users may face uncertain latency, which is a fatal blow to the benefits of the federated recommendation system.

Given the above scenario, we will consider the direction of federated cold-start recommendation to ensure recommendation services for new users. The authors in [43] have pioneered the exploration of the federated recommendation for cold-start items, where the number of users remains unchanged, but each user has new items to be recommended. However, this method does not apply to the federated recommendation for cold-start users due to the following additional challenges. Firstly, cold-start users need to match similar user information with warm clients, but user information is private and cannot be directly shared. Secondly, cold-start users may introduce new items, but it is challenging to identify which specific ones are new, thus aligning local items with items previously involved in federated training is not feasible. Lastly, cold-start users have not participated in federated training, and relying solely on transferred or local knowledge can lead to bias and fall short of achieving the expected performance.

To address these challenges, we in this paper investigate both a privacy-friendly and efficient **f**ederated **r**ecommendation framework for **c**old-**s**tart **u**sers dubbed FR-CSU that allows each cold-start user to gain improved recommendation with the adaptive fusion of transferred and local knowledge. More specifically, we first follow the hypothesis of sharing item embedding across all clients in a FedAvg manner while directly sharing user embedding with other clients is unacceptable due to privacy leakage. Inspired by [22], most cross-domain recommendation research transfers the common knowledge in the latent space between source and target domains to enhance recommendation performance. To transfer the aligned item knowledge in FR-CSU, we train a mapping function for each cold client to map the item embedding from the warm clients onto the cold clients. Since user information is private and cannot be easily transferred like

item information, warm clients in our method need to maintain a prototype network locally during training to generate privacy-friendly and transferable user information. This network will be aggregated using the FedAvg approach and backed up on the server after training. When cold-start users arrive, they can request this prototype network from the server to generate transferred user knowledge to assist in local recommendation tasks. Furthermore, we propose an adaptive knowledge fusion mechanism that initially integrates the transferred user and item knowledge using a linear function system and subsequently integrates the transferred knowledge with local knowledge to enhance local recommendations.

Through extensive experiments on various datasets with rating prediction and click-through rate, we show that FR-CSU significantly improves the recommendation performance compared to state-of-the-art approaches. The major contributions of this paper are summarized as follows:

- We are the first to study the problem of the federated recommendation for cold-start users. Different from the traditional federated recommendation, new users may emerge and require a better recommendation service. The balance between expense training overheads and ensuring the recommendation for all warm and cold users does matter in such a scenario.
- Then, to address this problem, we propose a novel federated recommendation framework for cold-start users named FR-CSU that can adaptively fuse transferred and local knowledge. To better transfer user and item knowledge separately, each cold-start user will train a local mapping function to transfer item knowledge. All warm clients maintain a user prototype network to provide privacy-friendly user information for cold-start users. An adaptive knowledge fusion mechanism is proposed to improve recommendations.
- Finally, extensive experiments are conducted on various datasets. Experimental results illustrate that our proposed model outperforms the state-of-the-art methods on both rating prediction and click-through rate tasks.

## 2 Related Work

**Cold-Start Recommendation.** Cold-start recommendation research focuses on providing high-quality recommendations for newly introduced items [13, 19, 28, 38, 45]. To tackle this challenge, several strategies have been devised. These include collaborative filtering [33, 36], content-based methods [8], and hybrid models [29]. Collaborative filtering analyzes past user interactions to find item similarities and common consumption patterns. Content-based methods, on the other hand, rely on item attributes to understand their features, allowing the system to assess correlations between new and existing items for more accurate recommendations. Hybrid models integrate both approaches by extracting relevant features from item attributes and incorporating them into the collaborative filtering framework, thus leveraging both methods' strengths to capture user interactions better. The authors in [43] recently combined the cold-start problem with the federated recommendation. In this work, each user will receive cold-start recommendations for newly added items, and our focus is on the federated recommendation for cold-start users.

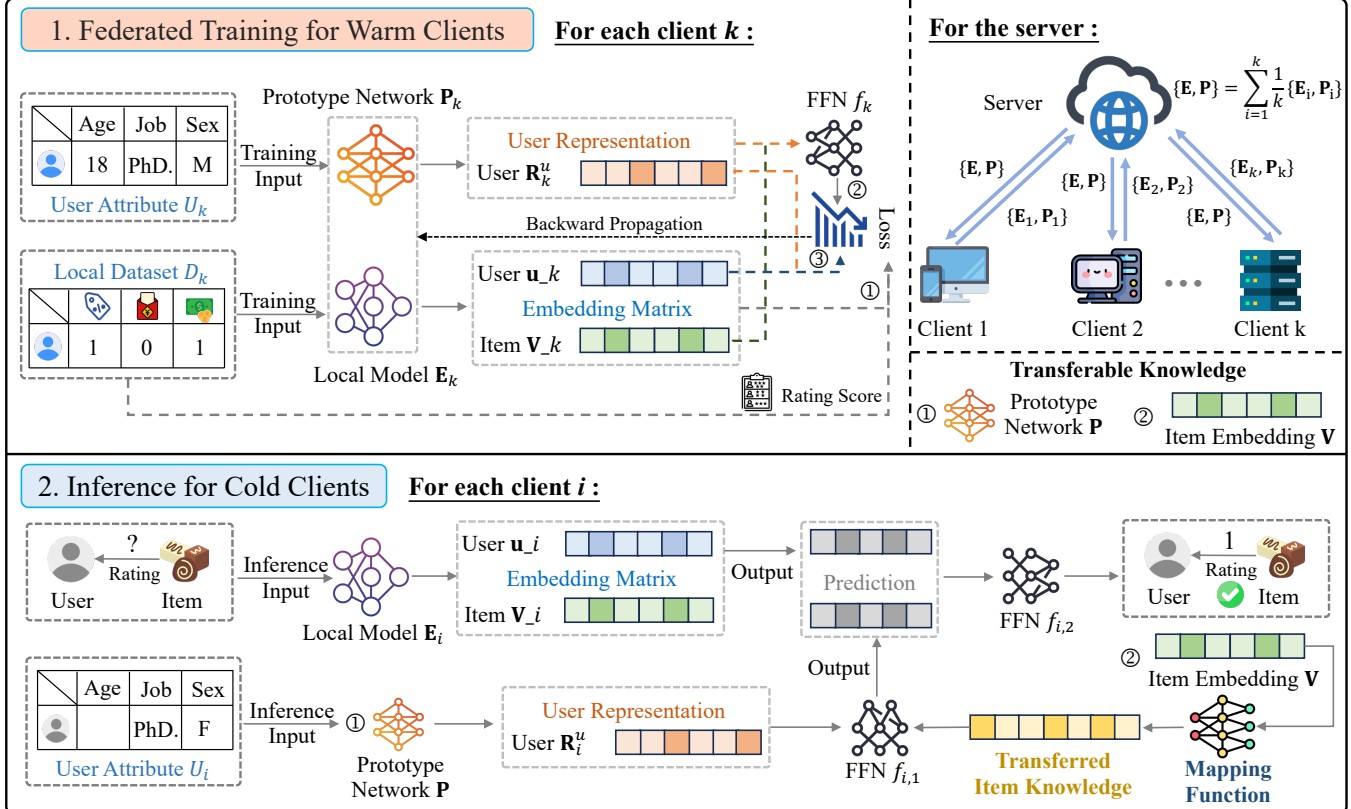

**Figure 2: The framework of FR-CSU.** *During the federated training for warm clients*, each client trains both the local embedding model and user prototype network with three alignment loss functions, which undergo global aggregation on the server. Upon completion, the server retains global item embeddings and the user prototype network as transferable knowledge. *During the inference for cold clients*, users initially download item embeddings and user prototype networks from established clients on the server. They then input local user attributes to derive transferred user representations. Ultimately, a linear function system (FFN) and collaborative filtering algorithms blend local and transferred knowledge, ensuring optimal recommendation performance.

**Federated Recommendation System.** Federated recommendation systems have recently attracted considerable attention due to increasing privacy concerns [16–18, 40]. Recent efforts have mainly focused on utilizing the interaction matrix, which is fundamental in basic recommendation scenarios [2, 44]. FCF [1], a pioneering collaborative filtering method under federated learning (FL), employs stochastic gradients for local model updates and FedAvg for global model aggregation. To safeguard user privacy, FedMF [3] integrates distributed matrix factorization into the FL framework, encrypting gradients before sending them to the server. Another distributed factorization approach, MetaMF [20], uses a meta-network to generate rating predictions and private item embeddings. FedPerGNN [37] allows users to maintain their graph neural network models, capturing high-order user-item relationships. However, in MetaMF and FedPerGNN, the server retains full model parameters, potentially revealing user interaction data and compromising privacy. FedNCF [12] adapts neural collaborative filtering to the federated setting, leveraging neural networks to learn complex user-item interactions, thereby enhancing model capabilities. Additionally, federated recommendation methods using diverse data sources have emerged,

considering multiple information streams. FedFast [24] enhances FedAvg with an active aggregation strategy to accelerate convergence, while Efficient-FedRec partitions the model into a server-side news model and a client-side user model, minimizing computational and communication overheads. Both approaches transcend the interaction matrix, incorporating user features and new attributes. pFedRec [21] offers a bipartite personalization mechanism for personalized recommendations. Building on this, FedRAP [15] balances global knowledge sharing and local personalization by applying an additive model to item embeddings and reduces communication costs through sparsity. Existing research primarily focuses on the traditional federated recommendation where the number of participating users is fixed. In this paper, we extend this scenario to a novel federated recommendation framework for cold-start users, effectively fusing transferable and local knowledge to improve recommendations.

## 3 Preliminary

**Traditional Federated Recommendation.** Before cold-start recommendation, it is necessary for all previous warm clients to collectively engage in federated recommendation training in order to

provide transferable knowledge for cold-start users. Here, we formulate the most general paradigm. Now we aim to collaboratively train a global model for $K$ total warm clients in FRS. We consider each client as a user $k$ with user information $u_k$. Each client $k$ has exclusive access to their private dataset $D_k = \{u_k, V_k, R_k\}$, where $V_k = \{v_k^1, v_k^2, \ldots, v_k^j\}$ contains $j$ items and $R_k = \{r_1, r_2, \ldots, r_j\}$ with $r_j \in \mathbf{R}$ representing user $k$'s interaction with item $v_j$. During the forward process, the original local data $\{u_k, V_k, R_k\}$ undergoes matrix factorization, transforming into $(\mathbf{u}_k, \mathbf{V}_k)$ via the embedding $\mathbf{E}$. The global dataset comprises the aggregation of all local datasets: $D = \{D_1, D_2, \ldots, D_K\} = \sum_{k=1}^{K} D_k$. The framework operates through the following steps:

(1) The central server initializes and distributes an untrained model to each client.

(2) Upon receiving the model, participating clients train it using their respective local data.

(3) These clients then upload their model parameters to the central server.

(4) The server consolidates the local models to update the global model, which is then dispatched to the clients for the next communication round.

Steps (2) to (4) form a complete communication cycle. This process repeats iteratively, with clients and the central server continuously exchanging information until the global model converges. The ultimate goal of the framework is to develop a global embedding model $\mathbf{E}$ that minimizes the cumulative empirical loss across the entire dataset $D$:

$$\min_{\mathbf{E}} \mathcal{L}(\mathbf{E}) := \sum_{k=1}^{K} \frac{|D_k|}{|D|} \mathcal{L}_k(\mathbf{E}). \tag{1}$$

$$\mathcal{L}_k(\mathbf{E}) = \begin{cases} \sum_{r_j \in R_k} \frac{1}{D_k} (r_j - \hat{r}_j)^2, \ \hat{r}_j = [\mathbf{u}_k]^T \mathbf{V}_k^j. \ ① \\ \sum_{r_j \in R_k} \frac{1}{D_k} - (r_j \log \hat{r}_j + (1 - r_j) \log(1 - \hat{r}_j)). \ ② \end{cases}$$

where $\mathcal{L}_k(\mathbf{E})$ is the loss in the $k$-th client. Here we define two loss functions for ① rating prediction and ② click-through rate tasks.

**Local Differential Privacy.** Local Differential Privacy (LDP) adds noise to gradients before sharing, thereby restricting the inferable information. The $(\epsilon, \delta)$-LDP definition is as follows:

**Definition 3.1.** A perturbation algorithm $M$ satisfies $(\epsilon, \delta)$-Local Differential Privacy if, for any adjacent datasets $D$ and $D'$, and for all possible output subsets $S$, the inequality below holds:

$$Pr[\mathcal{M}(D) \in S] \le e^{\epsilon} Pr[\mathcal{M}(D') \in S] + \delta \tag{2}$$

Here, $\epsilon$ represents the privacy budget of $M$, indicating the level of privacy protection, and $\delta$ is the probability of violating the privacy guarantee. A lower $\epsilon$ value signifies a narrower probability gap, hence stronger privacy.

## 4 Federated Recommendation for Cold-Start Users

In this section, we first outline the proposed method's overall framework and workflow algorithm. Next, we showcase the application of the inference phase for cold-start users. Lastly, we provide the privacy analysis for our method.

### 4.1 Framework and Workflow of FR-CSU

The key idea of FR-CSU is to adaptively transfer privacy-friendly user knowledge and item knowledge from warm clients to improve recommendation services for cold-start users with knowledge fusion. Our approach involves two main stages: distilling transferable knowledge from warm clients and fusing transferred and local knowledge to predict items for cold-start users. **During the learning stage for warm clients,** in addition to training local embedding models, each client needs to train a user prototype network and conduct global aggregation on the server. After training, the server retains the global item embeddings and user prototype network as transferable knowledge. **During the inference phase for cold-start users,** each user first downloads item embeddings and user prototype networks from the server and inputs local user information to generate transferred user representations. Finally, a linear function system and collaborative filtering algorithms are employed to integrate local and transferred knowledge to ensure recommendation performance. The workflow of the proposed framework is shown in Algorithm 1 and Figure 2 illustrates the FR-CSU framework.

### 4.2 Learning on the Warm Clients

To ensure that cold-start users can obtain effective transferred knowledge from the server, we first investigate federated recommendation training on warm clients. Previously, we have described the traditional federated recommendation training model in Section 3, where each client exchanges local knowledge by uploading embedding parameters. However, given that uploading user embeddings may leak user privacy, existing methods often choose to upload only item embeddings while keeping the user embeddings locally. Based on this, existing federated training methods with warm clients cannot provide relevant user information for cold-start users.

Inspired by [43], many cold-start recommendation tasks leverage additional item attribute information to assist in recommendations. In contrast, traditional recommendation tasks typically only require user-item interactions, utilizing collaborative filtering algorithms to achieve promising results. Without additional user attribute information, user information is learned solely through user embeddings. In cases where user embedding parameters are not uploaded, the server cannot obtain user information from the embedding. Therefore, we use user information from the user attribute as transferable user knowledge. To better learn and share user attribute information while protecting user privacy, we deploy a user prototype network for each user to learn a mapping from user attribute information to user representations. Supposed that the warm client $k$ possesses the user attribute information $U_k$ and the user prototype network $\mathbf{P}$, the local user representation $\mathbf{R}_k^u$ can be obtained by:

$$\mathbf{R}_k^u := \mathbf{P}(U_k), \ \text{where} \ \{\mathbf{R}_k^u, \mathbf{u}_k\} \in \mathbb{R}^{1 \times d}. \tag{3}$$

where $d$ represents the size of user embedding. Here, we utilize a user prototype network to map user attribute information into a user representation the same size as the user embedding to facilitate federated training and knowledge transfer. This user prototype network will achieve global aggregation to enable the communication of local

**Algorithm 1:** FR-CSU

**Input** : $T$: communication round; $K$: client number; $D_k$: local dataset for the client $k$; $\mathbf{u}_k$: local user embedding; $\mathbf{V}_k$: local item embedding; $\mathbf{R}_k^u$: local user representation; $U_k$: local user attribute information; $\mathbf{P}_k$: user prototype network; $\mathbf{M}_k$: local mapping function.

**Output** : $\{\mathbf{V}_k, \mathbf{u}_k\}$: embeddings of the cold-start user.

**1 Federated Training for Warm Clients:**

**2 for** $c = 1$ *to* $T$ **do**          // communication round

**3**     Server randomly selects a subset of devices $S_t$;

**4**     **for** *each selected client* $k \in S_t$ **in parallel do**

**5**        Generate the user representation $\mathbf{R}_k^u$ with (3);

**6**        Train the local embedding $\mathbf{V}_k$ and user prototype network $\mathbf{P}_k$ locally with (5);

**7**        Send the item embedding $\mathbf{V}_k$ and user prototype network $\mathbf{P}_k$ back to the server.

**8**     **end**

**9**     $\mathbf{V}, \mathbf{P} \leftarrow \text{ServerAggregation}(\{\mathbf{v}_k, \mathbf{P}_k\}_{k \in S_t})$

**10 end**

**11 Inference on Cold Clients:**

**12** Train the local embedding $\{\mathbf{u}_i, \mathbf{V}_i\}$ in a new format with (6);

**13** Download the item embedding $\mathbf{V}$ and user prototype network $\mathbf{P}$ from the server;

**14** Train the mapping function $\mathbf{M}_i$ with (7);

**15** Obtain the final prediction $\hat{r}_j$ by (8).

user knowledge:

$$\mathbf{R}^u := \frac{1}{K} \sum_{n=1}^{K} \mathbf{R}_n^u. \tag{4}$$

To better train the user prototype network and embedding model, we propose to achieve it by representation alignment. On the one hand, we need to align user embeddings with user representations to ensure more user information can be transferred. On the other hand, to align the consistency of transferred user knowledge between the inference stage and the learning stage (we will introduce the inference process for cold-start users in the next section), we additionally use a local linear layer $f_k$ to concatenate user representations and item embeddings for recommendation prediction. To ensure this, we employ the following alignment loss function for all warm clients:

$$\min_{\mathbf{E}, \mathbf{P}} \sum_{n=1}^{K} \left[ \mathcal{L}_n(\mathbf{E}) + \alpha \mathcal{L}(f_n(\mathbf{P}(U_n), \mathbf{V}_n), R_n) + \beta ||\mathbf{u}_n - \mathbf{P}(U_n)|| \right]. \tag{5}$$

where $\alpha$ and $\beta$ are hyper-parameters. The first loss term is the standard loss defined in (3). During training, each client only communicates the item embedding and user prototype network parameters with the server. After the training process, the server saves the final global item embedding $\mathbf{V}$ and the user prototype network $\mathbf{P}$ as the transferable knowledge.

## 4.3 Inference on the Cold Clients

After the federated training on warm clients is completed, cold-start users emerge and request high-quality recommendation services

from the server. Since cold-start users have not participated in federated training, we decided to integrate the knowledge migrated from warm clients with locally learned knowledge to enhance recommendations. Firstly, the cold client needs to train on its local dataset. To better transfer the knowledge from warm clients, we have also adjusted the local training process. Suppose the cold client $i$ needs to train its embedding model $\mathbf{E}_i$ and a linear function system on the local client, including two simple linear layers $f_{i,1}$ and $f_{i,2}$. The loss function is defined as follows:

$$\min_{\mathbf{E}_i, f_{i,1}, f_{i,2}} \mathcal{L}(f_{i,2}([\mathbf{u}_i]^T \mathbf{V}_i + \gamma f_{i,1}(\mathbf{u}_i, \mathbf{V}_i)), R_i). \tag{6}$$

where $\gamma$ is a hyperparameter. Motivated by the authors in [12] who repeatedly utilized embeddings for knowledge enhancement, we employ an additional pair of user embedding and item embedding. These embeddings are concatenated and fed into a linear layer. The output from this layer is then concatenated with the results from a collaborative filtering algorithm and input into the next linear layer to obtain the final recommendation prediction. Furthermore, this hyperparameter $\gamma$ can be determined by cold-start users to control the proportion of local knowledge and transferred knowledge. When $\gamma$ approaches 0, it indicates that cold-start users believe their local data is sufficient and high-quality, sufficient to achieve satisfactory recommendation results. Conversely, cold-start users will choose to utilize as much transferred knowledge as possible for local recommendations.

After local training, cold clients download the user prototype network $\mathbf{P}$ and item embeddings $\mathbf{V}$ from the server. Here, users can directly input their local user attribute information into the user prototype network $\mathbf{P}$ to obtain transferred user information. However, the transfer method for item embeddings remains to be considered. Cold-start users not only bring new user information but may also introduce new items not seen in federated training. Therefore, we first need to align the transferred item embeddings with local item information. To this end, we introduce a linear mapping function $\mathbf{M}_i$, commonly used in cross-domain recommendation to align item information between the source domain and the target domain [22]. We treat the transferred item embeddings as the input from the source domain and the local item embeddings as the target domain, aligning the item information accordingly:

$$\min_{\mathbf{M}_i} \mathcal{L}(\mathbf{M}_i(\mathbf{V}), \mathbf{V_i}). \tag{7}$$

Then, we incorporate the transferred knowledge with the local knowledge to enhance the recommendation:

$$\hat{r}_j = f_{i,2}\left([\mathbf{u}_i]^T \mathbf{V}_i^j + \gamma f_{i,1}(\mathbf{P}(\mathbf{u}_i), \mathbf{M}_i(\mathbf{V}_i^j))\right). \tag{8}$$

where $\hat{r}_r$ denotes the prediction result of the item $j$ for the user $i$. Thus, the cold-start user can enhance the local recommendation service by fusing the transferred and local knowledge by adjusting the $\gamma$ value.

## 4.4 Privacy Analysis

In our proposed FR-CSU framework, user embedding never leaves the local client. The only transferred knowledge is the user prototype network and item embedding, of which the item embedding is necessary and will not leak the user privacy alone. Compared to directly

transmitting user embeddings, the user prototype network in our method may also be attacked using gradient inversion or statistical correlation methods to obtain user representations. In recommendation systems, attackers often aim to get users' ground-truth ratings of items, which are calculated based on both user and item embeddings obtained through the attack. Assuming the attacker can access our user representations and item embeddings, it should be noted that our local loss function is not optimized through a simple collaborative filtering algorithm. As shown in Eq. (5), our loss function consists of three terms, including a local linear layer network and the utilization of user embeddings. Therefore, attackers cannot obtain users' rating preferences solely through user representations and item embeddings.

On this basis, we can further provide a Local Differential Privacy (LDP) [34] algorithm to protect our user prototype network from attacks. However, due to the relatively small number of parameters in the embedding, the recommendation performance decreases significantly when perturbation is applied to the model parameters to effectively protect user information. Considering that balancing privacy and recommendation performance is challenging, we abandon using the LDP algorithm in FR-CSU. The experiments regarding LDP will be analyzed in detail in Section 5. Moreover, embedding architectures are typically simple and vulnerable to attacks. The network architecture of the user prototype network is flexible and accessible, allowing us to adopt complex network structures, including attention mechanisms, to increase the difficulty of attacks and thereby enhance privacy protection.

## 5 Experiments

### 5.1 Datasets

A thorough experimental study has been conducted to assess the performance of the introduced FR-CSU in two popular scenarios with three recommendation datasets: (1) **Rating Prediction(RP)**: MovieLens-100K (ML-100K[1])[9] and MovieLens-1M (ML-1M[1])[9]. (2) **Click-Through Rate(CTR)**: QB-article[2][42]. These datasets are commonly employed for evaluating recommendation systems. Specifically, two MovieLens datasets sourced from the MovieLens platform feature movie ratings spanning 1 to 5, with each user contributing at least 20 ratings. When MovieLens datasets are used for the CTR task, we follow the setting in [15, 21] and make any rating higher than 0 in these datasets assigned 1. QB-article, an implicit feedback dataset, is derived from user interaction logs. The specific attributes of these datasets are outlined in Table 1.

### 5.2 Baselines

For a fair comparison with other works, we follow the protocols proposed by [23] to stimulate FL settings. We evaluate our method with the following baselines.

**1. Federated Recommendation:**
**FedMF** [3]: This method uses matrix factorization in a federated setting to reduce information leakage by encrypting user and item embedding gradients.
**FedNCF** [12]: This is the federated adaptation of NCF. It allows

---

[1]https://grouplens.org/datasets/movielens/
[2]https://github.com/yuangh-x/2022-NIPS-Tenrec

**Table 1: Experimental Details. Analysis of various considered settings of different datasets in the experiments section.**

| Attributes | Rating Prediction | | Click-Through Rate |
|---|---|---|---|
| | **ML-100K** | **ML-1M** | **QB-article** |
| Ratings | 100,000 | 1,000,209 | 134,990 |
| Users | 943 | 6,040 | 3,000 |
| Items | 1,682 | 3,952 | 5,144 |
| Sparsity | $s = 93.70\%$ | $s = 95.81\%$ | $s = 99.16\%$ |
| **For Warm Clients:** | | | |
| Total Users | 800 | 5000 | 2500 |
| Total Items | 1188 | 2584 | 4341 |
| Local training epoch | $E = 5$ | $E = 5$ | $E = 5$ |
| Communication Round | $T = 200$ | $T = 200$ | $T = 200$ |
| **For Cold Clients:** | | | |
| Total Users | 143 | 1040 | 500 |
| Total Items | 1297 | 3364 | 2357 |
| Local training epoch | $E = 5$ | $E = 5$ | $E = 5$ |

local updates of user embeddings on each client while synchronizing item embeddings on the server for global aggregation.
**FedMVMF** [7]: This is the federated version of matrix-vector multiplication matrix factorization. It enables local updates of user and matrix-vector features on each client while synchronizing the item embeddings and global parameters on the server.
**FedDCN** [32]: This is the federated adaptation of deep and cross networks. It allows local training of user-specific network layers on each client while synchronizing the shared cross-network and embedding layers on the server.
**FedWDR** [4]: This federated adaptation of wide and deep learning is tailored for recommendations. Clients train their wide and deep components to learn user-item patterns and relationships, leveraging the strengths of both wide and deep models while ensuring privacy.

**2. Federated Recommendation -w/Cold-Start:**
**FedPPR** [26]: This method allows clients to locally perform pairwise preference regression to estimate user preferences. The server then aggregates these local models for new users or items with limited interaction history.
**IFedNCF & IPFedRec [43]**: This method deploys a meta attribute network on the server to represent item attributes using raw item attributes and devises an item representation alignment mechanism for cold-start items.

### 5.3 Configurations

Unless otherwise mentioned, We set the number of clients to equal the number of users in the dataset. However, the notation "MovieLens-1M-3000" indicates that we randomly selected 3000 users from the original dataset for experimentation. During the training process for warm clients, we assume that 10% of clients will participate in each round of communication. In our main experiments, we used a three-layer linear network for the user prototype network in our method and a single-layer linear network for the mapping function. The hyperparameters are set as $\alpha = \beta = \gamma = 1$. Here, we report the

**Table 2: Performance comparison of various methods for the click-through rate (CTR) task. The best results are bold.**

| Categories | Methods | Metrics | MovieLens-100K | | | MovieLens-1M | | | QB-article-3000 | | |
|---|---|---|---|---|---|---|---|---|---|---|---|
| | | | @10 | @20 | @50 | @10 | @20 | @50 | @10 | @20 | @50 |
| FedRec | FedMF | HR | 0.1933 | 0.3249 | 0.5756 | 0.1686 | 0.3168 | 0.6169 | 0.1925 | 0.3282 | 0.5970 |
| | | NDCG | 0.0890 | 0.1221 | 0.1715 | 0.0744 | 0.1116 | 0.1706 | 0.0858 | 0.1198 | 0.1728 |
| | FedNCF | HR | 0.4264 | 0.6183 | 0.8678 | 0.4146 | 0.6258 | 0.7386 | 0.5757 | 0.6552 | 0.8317 |
| | | NDCG | 0.2046 | 0.2723 | 0.3187 | 0.1943 | 0.2205 | 0.3216 | 0.2869 | 0.3304 | 0.5543 |
| | FedMVMF | HR | 0.4213 | 0.6046 | 0.8174 | 0.2720 | 0.4273 | 0.6338 | 0.3051 | 0.4389 | 0.6896 |
| | | NDCG | 0.2015 | 0.2670 | 0.3119 | 0.1298 | 0.1667 | 0.2063 | 0.1785 | 0.2083 | 0.2567 |
| | FedDCN | HR | 0.5203 | 0.5955 | 0.7680 | 0.3907 | 0.5909 | 0.7315 | 0.6523 | 0.7497 | 0.8781 |
| | | NDCG | 0.1823 | 0.2103 | 0.5310 | 0.1804 | 0.2110 | 0.2912 | 0.3008 | 0.3356 | 0.4410 |
| | FedWDR | HR | 0.4253 | 0.6219 | 0.8777 | 0.3408 | 0.5178 | 0.7345 | 0.6118 | 0.7650 | 0.8993 |
| | | NDCG | 0.1931 | 0.2629 | 0.3240 | 0.1689 | 0.2135 | 0.2682 | 0.2950 | 0.3263 | 0.4493 |
| FedRec -w/Cold-Start | FedPPR | HR | 0.4148 | 0.6099 | 0.8895 | 0.4145 | 0.6131 | 0.7477 | 0.6073 | 0.7198 | 0.8782 |
| | | NDCG | 0.2043 | 0.2730 | 0.3251 | 0.1890 | 0.2192 | 0.2975 | 0.3091 | 0.3377 | 0.4291 |
| | IFedNCF | HR | 0.1469 | 0.2852 | 0.6386 | 0.1641 | 0.2931 | 0.6404 | 0.1729 | 0.2819 | 0.5514 |
| | | NDCG | 0.0668 | 0.0998 | 0.1678 | 0.0704 | 0.1022 | 0.1699 | 0.1077 | 0.1347 | 0.1992 |
| | IPFedRec | HR | 0.1352 | 0.3062 | 0.6827 | 0.1419 | 0.2700 | 0.5961 | 0.1852 | 0.3053 | 0.5739 |
| | | NDCG | 0.0616 | 0.0991 | 0.1733 | 0.0631 | 0.0953 | 0.1567 | 0.0958 | 0.1233 | 0.1775 |
| | FR-CSU (Ours) | HR | **0.6078** | **0.9223** | **0.9712** | **0.5430** | **0.7043** | **0.7552** | **0.6918** | **0.7749** | **0.9144** |
| | | NDCG | **0.2194** | **0.2868** | **0.6019** | **0.1976** | **0.2281** | **0.4971** | **0.3274** | **0.3466** | **0.6117** |

**Table 3: Performance comparison of various methods for rating prediction (RP). The best results are bold.**

| Task | Dataset | Metrics | FedRec | | | | | FedRec -w/Cold-Start | | | |
|---|---|---|---|---|---|---|---|---|---|---|---|
| | | | FedMF | FedNCF | FedMVMF | FedDCN | FedWDR | FedPPR | IFedNCF | IPFedRec | FR-CSU |
| Rating Prediction | MovieLens-100K | MAE | 2.9142 | 1.3686 | 3.2312 | 1.2202 | 1.1609 | 1.4702 | 1.1556 | 2.6735 | **0.9965** |
| | | RMSE | 3.2209 | 1.6734 | 3.4125 | 1.5337 | 1.4601 | 1.8606 | 1.4880 | 2.9488 | **1.3456** |
| | MovieLens-1M-3000 | MAE | 3.4450 | 1.7097 | 3.2268 | 1.3362 | 1.2680 | 1.5221 | 1.2810 | 3.2806 | **1.0853** |
| | | RMSE | 3.6462 | 2.0393 | 3.4370 | 1.6059 | 1.5815 | 1.8856 | 1.6327 | 3.5102 | **1.4744** |
| | MovieLens-1M | MAE | 3.3460 | 1.2593 | 3.2682 | 1.1684 | 1.2350 | 1.3393 | 1.1545 | 3.5514 | **0.9709** |
| | | RMSE | 3.5367 | 1.5519 | 3.4551 | 1.4923 | 1.4690 | 1.7042 | 1.5251 | 3.7277 | **1.3218** |

Mean Absolute Error (MAE) and Root Mean Square Error (RMSE) as the metrics for the rating perdition [22] and Hit Rate (HR@K) and Normalized Discounted Cumulative Gain (NDCG@K) for the click-through rate [15, 21]. We illustrate all the settings with all the benchmark parameters in Table 1. Each experiment set is run twice, and we take each run's final ten rounds' accuracy and calculate the average value for all cold-start users. We use Adam as an optimizer with a linear learning rate schedule. We set the remaining parameters according to the values in the original open-source code.

## 5.4 Performance Overview

**Main Results.** Table 2 and 3 comprehensively showcase the efficacy of various methods when tasked with rating prediction and click-through rate estimation. Notably, our proposed method, FR-CSU, demonstrates **superior** performance across multiple datasets and metrics. For the CTR task, when examining the HR@10 metric, FR-CSU outperforms other methods by significant margins. As

the evaluation metrics extend to higher positions (e.g., @20, @50), FR-CSU continues to exhibit robust performance, indicating its ability to provide relevant recommendations even when considering a larger pool of candidates. For the rating prediction task, FR-CSU also performs well, but we observe that the methods designed for the cold-start problems can achieve a much better result than those for the CTR task. Overall, the experimental results demonstrate the effectiveness and robustness of our proposed method, FR-CSU, in both the CTR task and rating prediction task. FR-CSU's superior performance across multiple datasets and metrics, as well as its ability to handle the cold-start problem, make it a promising approach for federated recommender systems.

**Ablation Study.** As shown in Table 4, we evaluate the effects of each module in our model via ablation studies. -w/o mapping item and -w/o user representation denote the performance of cold-start users without using the transferred user knowledge or aligned item

**Table 4: Ablation study of FR-CSU two federated recommendation tasks on MovieLens-100K.**

| Dataset | Method | MovieLens-100K | | | |
|---|---|---|---|---|---|
| | | HR@10 | NDCG@10 | MAE | RMSE |
| ML-100K | FR-CSU | **0.6220** | **0.2239** | **0.9965** | **1.3456** |
| | -w/o mapping item | 0.6066 | 0.2175 | 1.1881 | 1.4793 |
| | -w/o user representation | 0.5326 | 0.2125 | 1.2022 | 1.4947 |

embedding for inference. Compared with FR-CSU, the performance of FR-CSU -w/o user representation degrades evidently. Specifically, the relatively less prominent role of the mapping function module may be constrained by issues related to the dataset, specifically that our new users did not introduce many new items or items that are sparse among warm clients. Experiment results verify the effectiveness of all modules, confirming all modules are essential to train a robust federated recommendation model for cold-start users.

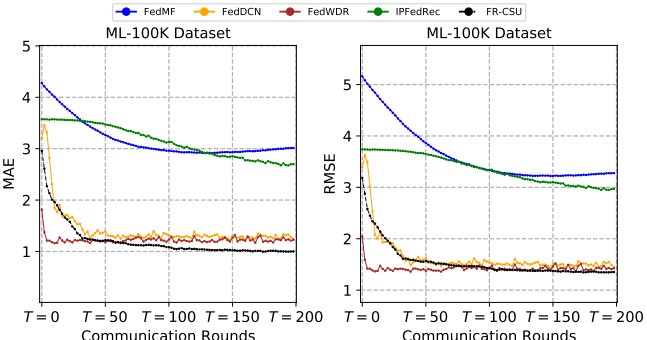

**Figure 3: Convergence and efficiency comparison of various methods on the ML-100K dataset.**

**Communication Efficiency.** Fig. 3 presents a comprehensive evaluation of various methods, focusing on both convergence and communication efficiency. In this assessment, we record the performance of these methods every two iterations. Even though the training methodology employed by FR-CSU diverges from other established baselines, it maintains a commendable convergence rate. This is a testament to the robustness and adaptability of the FR-CSU approach. Furthermore, the incorporation of additional user attribute information catalyzes faster convergence. This enhancement not only underscores the significance of leveraging rich, supplementary data but also highlights the innovative approach taken by FR-CSU in integrating such information seamlessly into its training process.

**Parameter Sensitivity Analysis.** Fig. 4 provides the two metrics under different ratios between warm and cold clients. In this figure, FR-CSU performs best with different ratios, and the lower MAE and RMSE loss is achieved by raising the ratio $\beta$, which means more warm clients and fewer cold-start users. It has been proven that when there are sufficient warm clients that can provide richer transferable knowledge, cold-start users indeed do not need to undergo federated training to achieve promising performance. Conversely, the server can consider incremental learning methods to address recommendation scenarios involving many new users. Since this paper focuses

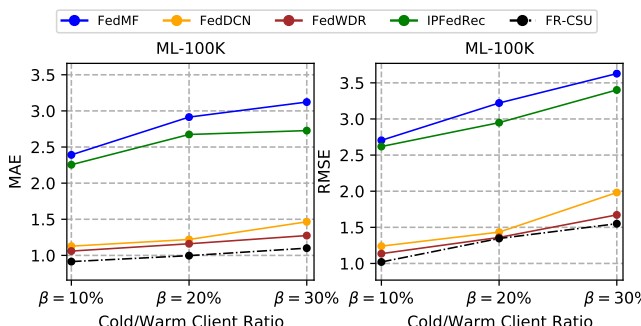

**Figure 4: Performance comparison of various methods w.r.t. ratio $r$ between warm clients and cold clients.**

**Table 5: Evaluation of differential privacy with the user embedding and prototype network on MovieLens-100K.**

| Dataset | Method | Metric | Privacy Budget $\epsilon$ | | | | |
|---|---|---|---|---|---|---|---|
| | | | $\epsilon = 0.1$ | $\epsilon = 0.2$ | $\epsilon = 0.3$ | $\epsilon = 0.4$ | $\epsilon \geq 0.5$ |
| ML-100K | -w/user embedding | MAE | 2.1165 | 1.9703 | 1.7456 | 1.5473 | 1.4527 |
| | | RMSE | 2.6750 | 2.3495 | 2.1435 | 1.9717 | 1.8000 |
| | FR-CSU | MAE | 3.3757 | 3.0541 | 2.6976 | 2.3714 | 2.1350 |
| | | RMSE | 3.5980 | 3.2512 | 2.9122 | 2.6066 | 2.3820 |

on federated recommendation scenarios where warm clients are the majority, related work on incremental learning can be carried out in our future research.

**Privacy Analysis.** We will explore the feasibility of the classic privacy protection method, the LDP algorithm. Here, we consider two scenarios: (1) Directly transferring user embeddings protected by LDP without using a user prototype network; (2) Adding LDP protection to the user prototype network in FR-CSU. We use the Laplacian noise and set the strength from 0.1 to 0.5 with an interval of 0.1. As shown in Table 5, the model performance degrades as the noise strength increases. However, when we add noise to the embedding, it directly interferes with the transferred user information, resulting in a significant drop in model performance. On the other hand, adding noise to the user prototype network would also cause a relatively slighter decline in model performance. Nevertheless, considering the challenges in setting noise strength and balancing recommendation performance and user privacy in the LDP method, we do not incorporate LDP as part of the FR-CSU method.

## 6 Conclusion

In this paper, we delve into the challenges of developing a federated recommendation system tailored for cold-start users where new users ask for a better recommendation service for local items with the transferred knowledge from converged warm clients. We propose a privacy-friendly yet effective framework, FR-CSU, which transfers both user and item knowledge from warm clients and allows cold-start users to fuse the transferred knowledge with the local knowledge adaptively to provide a personalized recommendation. Extensive experiments conducted on various settings and baselines show that FR-CSU significantly improves recommendations.

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
