# OpenReview forum: "Personalized Federated Recommendation for Cold-Start Users via Adaptive Knowledge Fusion"
_ACM.org/TheWebConf/2025/Conference — WWW 2025 Poster_

### Official Review · Reviewer_X3HN · 2024-11-29

**Novelty:** 5
**Technical Quality:** 5

**Review:**

This paper studies the problem of cold-start federated recommendation, where new users join the training intermediately. The authors propose FR-CSU to transfer knowledge from warm users to new comers, while keeping the user embeddings local. Similar to cross-domain recommendation, the authors leverage bridge items to transfer. Experiments are conducted on multiple real-world datasets.

Pros

1. The scenario of cold-start federated recommendation is interesting and of great importance in practical applications.
2. The paper is well-structured and comfortable to read.

Cons

1. This paper is similar to [1] in terms of its technical framework, experiments, and even writing style. The main difference is that the authors introduce a prototype network to incorporate information for new users. Please clarify any other technical contributions if I have missed.
2. Why can a prototype network trained on warm users effectively predict the information for new users?
3. Other than the main experiments, most experiments are conducted on ML-100k dataset, which is relatively simple (a toy example). And no supplementary result is reported in the appendix.
4. Experiments are recommended to run multiple times than twice to ensure the reliability of the findings.

[1] When Federated Recommendation Meets Cold-Start Problem: Separating Item Attributes and User Interactions, WWW'24.

**Questions:**

1. Why did you choose to build the ML-1M-3000 dataset? Why not use other versions of the ML dataset, such as ML-10M or ML-20M?

**Reviewer Confidence:**

3: The reviewer is confident but not certain that the evaluation is correct

**Scope:**

3: The work is somewhat relevant to the Web and to the track, and is of narrow interest to a sub-community

---

### Official Review · Reviewer_K28p · 2024-11-29

**Novelty:** 5
**Technical Quality:** 5

**Review:**

This paper reports FR-CSU, which addresses federated recommendation for cold-start users by enabling high-quality recommendations without retraining the entire model. It combines local data with knowledge transferred from warm clients, tackling shifts in item embeddings and ensuring privacy. Key innovations include a local mapping function for item alignment, a privacy-friendly user prototype network, and a system to fuse transferred and local knowledge. Experiments show FR-CSU outperforms state-of-the-art methods while being efficient and privacy-preserving.

Pros:

1. The cold-start problem in federated recommendation is an interesting problem.
2. The reported results seem promising.
3. This paper is easy to read.

Cons:

1. The coverage of FedRS methods for comparison in experiments is limited.

2. Since the useful information from cold-start users is limited, I wonder whether it is sufficient to select the useful knowledge from the active users. In other words, it may cause echo-chamber effect.

3. Only two datasets are used for experiments. There is not description of these datasets. In addition, there is no introduction to the reason of selecting these datasets.

**Questions:**

1. Can you include more FedRS methods in experiments?

2. Do you think the limited information from cold users is sufficient to select the useful global knowledge? Does your selection method cause echo-chamber?

3. Why do you select these two datasets?

**Reviewer Confidence:**

2: The reviewer is willing to defend the evaluation, but it is likely that the reviewer did not understand parts of the paper

**Scope:**

3: The work is somewhat relevant to the Web and to the track, and is of narrow interest to a sub-community

---

### Official Review · Reviewer_3xQy · 2024-12-01

**Novelty:** 5
**Technical Quality:** 5

**Review:**

This paper addresses the cold-start problem for users in federated settings and proposes a new framework, FR-CSU. This framework utilizes an adaptive knowledge fusion mechanism to transfer knowledge from warm users to cold users, enabling federated cold-start solutions specifically for users.

The paper clearly identifies the challenges posed by cold-start users, particularly in the context of new items that may be introduced for new users. The design of user prototype networks and the corresponding transfer strategy are well-motivated and reasonable. Overall, the research quality is high.

While the structure of the paper is well-organized, some sections are not sufficiently detailed. For example, the explanation of the three alignment mechanisms for each client and the different networks employed by each client is not easy to follow.
The paper draws inspiration from IFedRec and modifies the original server-side project feature learning to client-side user feature learning. Based on this, the authors design a knowledge transfer strategy. This is the first federated framework to address user cold-start problems, demonstrating a strong level of innovation.

The paper has two strengths:(1)The user prototype network learns features from warm users and transfers them to cold users, thereby protecting privacy.(2)The architecture diagram is clear and aesthetically appealing.
But it exists some shortcomings:(1)In the warm-user learning phase, the paper only describes the overall loss function for each client, but does not provide details about the individual networks, such as the  user prototype network and the FFN.(2)In the cold-user inference phase, the description of the two FFNs and the local model is insufficient.

**Questions:**

(1)Warm User Training Phase: The paper introduces the addition of the outputs from the other two networks (user prototype network and FFN) as regularization terms in the recommendation model's loss function (Eq5). However, it does not explain how these two networks (the user prototype network and FFN) are trained, their architecture, loss functions, or their roles. Based on the cited IFedRec, I speculate that the user prototype network serves to align user representations and user embeddings, but the role and training process of the FFN are unclear.

(2)Cold User Inference Phase: During the cold user inference stage, the cold user trains the recommendation model and the two FFNs locally. Given that cold users typically have limited or no data, how are they trained? How is the embedding model for cold users obtained? If sufficient data were available, wouldn’t this become an inductive learning task?

(3)This paper is designed for cold-start users, but IFedRec is designed for cold-start items, so I think it is more reasonable to compare it to the inductive method rather than IFedRec if clients have sufficient data.

**Reviewer Confidence:**

3: The reviewer is confident but not certain that the evaluation is correct

**Scope:**

4: The work is relevant to the Web and to the track, and is of broad interest to the community

---

### Official Review · Reviewer_jKqp · 2024-12-01

**Novelty:** 4
**Technical Quality:** 4

**Review:**

Summary：
The paper proposes FR-CSU, a federated recommendation framework for cold-start users, focusing on knowledge transfer from warm clients while preserving privacy. The work addresses a challenge in federated recommendation systems and provides a novel solution through user prototype networks and adaptive knowledge fusion.

Pros:
1. The paper clearly identifies the challenges faced by cold-start users in federated settings in Section 1 and provides a strong theoretical justification for why retraining the entire system is impractical.
2. The authors demonstrate the applicability and superiority of the FR-CSU method through experiments on multiple datasets. The experimental results show that the method outperforms state-of-the-art approaches on several benchmark datasets.

Cons:
1. While Section 5.4 demonstrates the convergence trends through Figure 3, there is no investigation of how different training configurations (client numbers, participation rates) affect communication overhead, nor analysis of the relationship between model scale and communication costs. The paper also fails to address potential system bottlenecks in heterogeneous device environments.
2. In Section 4.3, the paper introduces parameter γ to balance local and transferred knowledge fusion in equation (6). Although the theoretical meaning of γ is clearly explained, there is no analysis of how different γ values impact model performance, no discussion of adaptive adjustment strategies for γ when facing data distribution shifts, and no guidelines for parameter selection in different scenarios (e.g., high proportion of new items).

**Questions:**

Q1: In Section 4.3, the paper uses a mapping function \( M_i \) to align item embeddings for cold-start users. How does this mapping mechanism perform when cold-start users introduce significantly different items? Is the method still effective when there is a large disparity in item distribution between cold-start and warm users?

Q2: The paper mentions that the parameter \( \gamma \) in the knowledge fusion mechanism is set based on cold-start users’ assessment of their local data quality. Could an automatic method based on data characteristics be developed to select the optimal \( \gamma \)?

**Reviewer Confidence:**

3: The reviewer is confident but not certain that the evaluation is correct

**Scope:**

4: The work is relevant to the Web and to the track, and is of broad interest to the community

---

### Official Review · Reviewer_AgXj · 2024-12-02

**Novelty:** 5
**Technical Quality:** 5

**Review:**

To address the problem of providing effective recommendations for cold-start users in federated systems, this paper introduces a novel method named FR-CSU (Federated Recommendation for Cold-Start Users). It leverages techniques such as adaptive knowledge fusion, local mapping functions, and prototype networks to tackle the challenges of privacy-preserving knowledge transfer and sparse data alignment. Experimental results on multiple datasets demonstrate the effectiveness of the proposed method.

Pros:

1. The research problem is meaningful. The user cold-start problem in the context of federated recommendation has been relatively underexplored, making this study timely and valuable.

2. The proposed method, FR-CSU, is well-designed and incorporates solutions for knowledge transfer while preserving user privacy.

3. The experimental results effectively validate the performance of the proposed method。

Cons:

1. For the CTR task, while HR and NDCG are reported, a more common evaluation metric, accuracy, is omitted, which may limit the comprehensiveness of the results.

2. The lack of provided source code restricts the reproducibility of the research.

**Questions:**

1. Why was accuracy not included as an evaluation metric for the CTR task?

**Reviewer Confidence:**

4: The reviewer is certain that the evaluation is correct and very familiar with the relevant literature

**Scope:**

4: The work is relevant to the Web and to the track, and is of broad interest to the community